# Validated Tools for Assessing Anxiety and Depression in Nurses: A Systematic Review

**DOI:** 10.3390/ijerph22111714

**Published:** 2025-11-13

**Authors:** Gabriel Reyes Rodríguez, Leticia Cuellar-Pompa, Natalia Rodríguez Novo, Miguel López Martínez, José Ángel Rodríguez Gómez

**Affiliations:** 1Nursing Department, Hospital Universitario Nuestra Señora de la Candelaria, 38010 Santa Cruz de Tenerife, Spain; 2Instituto de Investigación en Cuidados del Colegio Oficialde Enfermería de Santa Cruz de Tenerife, 38010 Santa Cruz de Tenerife, Spain; 3Nursing Department, Faculty of Health Sciences, Universdad de La Laguna, 38200 La Laguna, Spain; 4Emergency Department, Hospital Universitario Nuestra Señora de Candelaria, 38010 Santa Cruz de Tenerife, Spain; 5Nursing Department, Faculty of Health Science, University of La Laguna, 38200 La Laguna, Spain

**Keywords:** anxiety, depression, nurses, mental health, assessment instruments, screening tools, health personnel, nursing, quality of life

## Abstract

Background: Nurses experience substantial anxiety and depression; robust, validated instruments are needed. We aimed to identify tools used to assess these conditions in nurses. Methods: A systematic review was conducted in December 2024 and registered in OSF and PROSPERO. MEDLINE, Embase, CINAHL, and PsycINFO were searched for quantitative studies (2014–2024) in English/Spanish that included nurses only and used standardized measures. Two reviewers screened and extracted the data; quality was appraised with JBI checklists, narrative synthesis only. Results: Twenty-two studies (n = 10,710 nurses) met the criteria. Most were cross-sectional with non-probability sampling; the overall risk of bias was moderate in 19 studies and high in 3. The most frequently used instruments were PHQ-9, GAD-7, GHQ-28, and BDI; across versions, PHQ (PHQ-2/PHQ-9) predominated. Heterogeneity precluded meta-analysis. Discussion: The available tools support routine screening in nursing populations, but reliance on self-reports and scarce formal cross-cultural validation in practicing nurses limit inference and generalizability. Conclusions: Screening programs in nursing should pair brief self-report instruments with objective indicators and standardized protocols; future studies should prioritize contextualized validation and robust longitudinal designs.

## 1. Introduction

The mental health of healthcare workers is now a global priority, especially in the wake of the COVID-19 pandemic. Anxiety and depression, among the most common disorders worldwide, have increased dramatically: according to the World Health Organization (WHO), their combined prevalence increased by approximately 25% during the first year of the pandemic [1]. This phenomenon particularly affects nursing staff. Several studies report that nurses face high levels of chronic work-related stress [2], attributed to work overload, long hours, and other factors, which impairs their well-being and even the quality of patient care [3,4]. Consequently, anxiety and depression have become common problems among this group.

International evidence confirms the severity of this situation. A global meta-analysis found that approximately one in four female healthcare professionals experienced symptoms of anxiety (23.2%) or depression (22.8%) during the pandemic [5], with significantly higher rates among nurses (as well as female staff in general) compared to their male counterparts or physicians [6]. Similarly, a large international survey conducted in 35 countries revealed that between 23% and 61% of nurses have experienced work-related anxiety or depression symptoms [7]. Although rates vary across regions, these findings confirm that mental health issues in nursing are both a global and significant problem.

The repercussions of this problem are significant at both the individual and institutional levels. Prolonged psychological distress in nurses has been associated with a higher risk of burnout, a decrease in the quality of care, and even an increase in the intention to leave the profession. In Europe, for example, it has been observed that work–life imbalance, coupled with precarious working conditions, significantly increases the likelihood that nursing staff will consider leaving clinical practice. This situation not only affects the health of nursing professionals but also exacerbates the nursing staff shortage, jeopardizing the sustainability of healthcare systems.

Given this situation, early detection of anxiety and depression in nursing staff is key to activating appropriate support measures. Given the high prevalence of these disorders in the profession, it is essential to use accurate and validated assessment instruments that facilitate their early identification and inform the implementation of effective preventive strategies [8]. Conversely, inadequate assessment carries the risk of underestimating the true magnitude of the problem, which could translate into a lack of institutional support and insufficient preventive interventions.

Various scales are available in clinical practice to measure anxiety and depression; however, not all have been specifically designed for nurses or validated in specific contexts. Even though related constructs such as burnout, stress, or insomnia are also relevant to mental health, they represent distinct phenomena with different diagnostic frameworks.

Although recent studies have used various instruments, there is still no consensus on which is most appropriate in this field. Therefore, it is necessary to identify which of these tools have solid psychometric support and are suitable for assessing nurses’ mental health.

The findings may be valuable for international healthcare settings that share cultural similarities. Identifying tools with adequate psychometric properties will guide professional practitioners and researchers in the selection or adaptation of instruments to assess anxiety and depression in nurses, thus contributing to improved detection and management of these conditions.

The objective of this review is to identify the validated instruments most frequently used to assess anxiety and depression in nurses and to describe their main characteristics and context of use.

## 2. Methods

A systematic review was conducted following the PRISMA 2020 statement [9]. Quality/risk of bias assessment was performed using the JBI checklists [10], applying the corresponding tool according to the design (cross-sectional, quasi-experimental, and cohort).

### 2.1. Protocol and Registration

The protocol was prospectively registered with the Open Science Framework (OSF), which hosts the complete search strategy, terms used, extraction templates, and the database with all search files: https://osf.io/r9cmb/?view_only=14d2e77fb7b54062933be1cfd364b80a (accessed on 21 February 2025). The protocol was also registered with PROSPERO; the registration form can be found at the following link: https://www.crd.york.ac.uk/PROSPERO/view/CRD420251017806 (accessed on 23 March 2025).

### 2.2. Eligibility Criteria

Inclusion Criteria: Original studies with a quantitative design, published between 2014 and 2024 (inclusive), in English or Spanish, which used standardized instruments to assess anxiety and/or depression in nursing professionals were included. Only studies with a sample consisting entirely of nursing staff were selected.

Exclusion Criteria: Studies that did not use standardized instruments to assess anxiety or depression, those that did not explicitly identify the inclusion of nursing staff, and those not available in full text were excluded. Editorials, letters to the editor, and reviews were also excluded.

Grey literature and non-English/Spanish studies were excluded to ensure methodological rigor, data reproducibility, and comparability of validated instruments.

### 2.3. Information Sources

An initial manual search was conducted in December 2024 using PROSPERO (Centre for Reviews and Dissemination, University of York), the TRIP meta-search engine, and Google Scholar to identify previous high-quality systematic reviews and primary studies relevant to the topic of interest.

The final search strategy was performed between December 17 and 23, 2024, in the following databases: MEDLINE and PreMEDLINE via Ovid, Embase via Elsevier, CINAHL Complete, and APA PsycINFO via EBSCOhost.

### 2.4. Search Strategy

Relevant keywords were identified using the snowball technique from the titles and abstracts of the studies consulted in the exploratory phase of this review. MeSH/Emtree searches were identified in the thesauri of each database. The structured search strategy was initially designed and tested in Embase, using different iterative combinations to maximize both precision and recall. Saturation was considered reached when new combinations no longer yielded additional relevant studies. The final strategy was subsequently adapted to the syntax of each platform. The strategies were evaluated using the PRESS (Peer Review of Electronic Search Strategies) protocol [11], ensuring their conceptual validity, as well as the appropriate use of vocabulary and syntax.

Finally, the reference lists of the included studies were manually reviewed to identify other potentially relevant publications. The results were exported to the Rayyan platform [12], where duplicates were removed and the selection process was carried out.

### 2.5. Study Selection Process

The study selection was conducted in two phases using the Rayyan platform. Two independent reviewers screened titles and abstracts, followed by a full-text evaluation of potentially eligible studies, applying the previously established eligibility criteria. A training session with 25 randomly selected references was carried out to ensure consistency, achieving 75% agreement. Discrepancies were resolved by consensus, and a third reviewer was not required as per the registered protocol. Reasons for exclusion were recorded at each stage in accordance with PRISMA recommendations, and the entire process was documented using a PRISMA flowchart.

### 2.6. Data Extraction Process

Data extraction was performed using a structured template that included author, year, country, study design, sample size, tool used, and main research results. One reviewer extracted the information, and a second reviewer verified it.

The protocol contemplated prioritizing instruments validated in Spanish; however, the heterogeneity of the identified studies (language, context, and reporting of measurement properties) prevented this criterion from being systematically applied. Therefore, before extraction, the scope was expanded to include international studies that used standardized nursing instruments; this modification was recorded in the project documentation (OSF/PROSPERO).

### 2.7. Summary of Results

Given the heterogeneity of designs, contexts, and metrics, a narrative synthesis was performed without meta-analysis. Methodological quality was assessed using the JBI checklists by design. Two reviewers independently assessed the quality and resolved discrepancies by consensus. An overall judgment (low/moderate/high) was given considering: inclusion criteria, description of the setting, measurement of exposure/condition, identification and control of confounders, validity/reliability of the outcome instruments, and adequacy of the statistical analysis. A narrative synthesis of each instrument’s advantages and limitations was developed based on the information extracted from the included studies. The process was conducted jointly by the reviewers through discussion and consensus, focusing on aspects such as diagnostic sensitivity, brevity, clinical applicability, and overall suitability for nursing populations.

### 2.8. Additional Considerations Regarding the Methodological Approach

The present study aimed to identify assessment tools already available and applicable to nursing staff. Although the protocol prioritized instruments validated in Spanish, the methodological and linguistic heterogeneity of the evidence made it impossible to maintain this filter uniformly; therefore, international studies were included. This decision broadened the perspective and allowed for contextualizing the use of the instruments at a global level, without conducting systematic comparisons or assessing their cross-cultural appropriateness. The methodology was designed to ensure rigor and transparency in the selection and to maximize the practical utility of the results for future nursing research.

All methodological steps were carried out in accordance with the PRISMA 2020 statement to ensure transparency and reproducibility.

## 3. Results

The literature search identified a total of 2193 records, of which 1977 were evaluated after eliminating duplicates. Following a full-text review, 22 studies that met the eligibility criteria were finally included. These studies provided information on the use of validated tools for assessing staff anxiety and depression, involving a total sample of 10,710 professionals from different countries. The search process is shown in the flow diagram in Figure 1, proposed by PRISMA [13].

Data were collected from a total of 10,710 nurses from the following countries: China, Australia, Bangladesh, Portugal, Germany, Japan, the United States, Poland, Iran, Turkey, the Philippines, Spain, Hungary, Italy, Slovakia, the Czech Republic, and Malaysia. The year 2021 had the highest number of publications.

Table 1 provides a detailed summary of the studies included in this systematic review, outlining their key methodological anRFd contextual characteristics. For each study, information is presented on the author, year of publication, study design, sample size, country, and the specific instruments used to assess anxiety and/or depression. It also summarizes the main outcomes reported in relation to these constructs, offering an overview of how each instrument has been applied within different nursing contexts.

As a result of the systematic review, the most commonly used tools were found to be the PHQ-9, GAD-7, GHQ-28, and BDI, each of which was used in four different studies. However, when considering the set of tools (without distinguishing between versions), the PHQ was the most widely used, with six mentions, both in its abbreviated (PHQ-2) and full (PHQ-9) versions, establishing itself as the most widely used instrument.

In the field of depression assessment, several validated tools were identified. Among them, the Patient Health Questionnaire (PHQ), in its PHQ-9 [14,15,18] and PHQ-2 [16,17] versions. The PHQ-9 is designed to identify the diagnostic criteria for depression and stands out for its high sensitivity and specificity, making it ideal for primary care and populations with disabilities. The PHQ-2 consists of two items and is used as an initial rapid screening, with the recommendation to expand the assessment with the PHQ-9 in the event of positive results.

Another widely used tool was the Beck Depression Inventory (BDI) [26,27,28,29], which includes 21 items and allows for classifying the severity of depressive symptoms from mild to severe. Likewise, the CESD-R-10 [35], a shortened version of the Center for Epidemiologic Studies Depression Scale, was presented as a practical alternative in large-scale studies due to its brevity and simplicity.

Regarding anxiety, the Generalized Anxiety Disorder or GAD-7 [17,19,20] was one of the most widely used instruments. This self-administered, seven-item questionnaire has demonstrated high validity and reliability in the assessment of generalized anxiety disorder. The GAD-2 [16] was identified for rapid screening settings, which includes the first two items of the GAD-7, although with lower diagnostic accuracy.

The Beck Anxiety Inventory (BAI) [34] and the State–Trait Anxiety Inventory (STAI) [22] complement the repertoire. The BAI assesses emotional, cognitive, and physical symptoms associated with anxiety through 21 items, while the STAI allows for differentiation between transient (state) and chronic (trait) anxiety.

The General Health Questionnaire (GHQ) was also identified, in its GHQ-12 [21] and GHQ-28 [22,23,24,25] versions. The GHQ-12 version is appreciated for its applicability in population studies, while the GHQ-28, with its specific subscales, allows for a more detailed assessment of somatization, anxiety and insomnia, social dysfunction, and major depression.

Likewise, the Hospital Anxiety and Depression Scale (HADS) [33], designed for the hospital setting, excludes somatic symptoms that could interfere with the assessment and classifies anxiety and depression by severity level.

Finally, the Depression, Anxiety, and Stress Scale (DASS-21) [30,31,32], which consists of three 7-item subscales, allows for the joint assessment of depression, anxiety, and stress. Its scores are multiplied by two to estimate the severity of each domain, making it a versatile and useful tool in multicultural contexts.

Lastly, a comparative table was prepared summarizing the main strengths and limitations of the identified assessment tools. This summary facilitates comparison between scales based on criteria such as diagnostic sensitivity, brevity, clinical applicability, and appropriateness to the healthcare context. Table 2, therefore, provides a structured view of the key elements that can guide the selection of the most appropriate instrument according to the objectives of each study or intervention.

### Methodological Quality of the Included Studies

Twenty-two studies were evaluated. Overall, 19 presented a moderate risk of bias and three a high risk (mainly non-randomized pre–post studies without a control group). Cross-sectional designs with non-probability sampling and self-report measures predominated. Several studies performed multivariate adjustment, and one applied propensity score matching in addition to regression; however, their cross-sectional nature and the use of self-reports limit causal inference and increase the risk of information bias.

Table 3 summarizes the overall quality assessment and the main limitations per study, while Appendix A presents the full appraisal (JBI criteria, sampling, control of confounders, analysis, and notes) and is available as Appendix A.

Taken together, these findings reveal a predominance of studies with moderate methodological rigor, reflecting common challenges in nursing research such as limited randomization, convenience sampling, and reliance on self-report tools. While these designs provide valuable descriptive evidence, they hinder the establishment of causal relationships and reduce comparability across settings. The absence of longitudinal data further restricts the capacity to evaluate temporal or directional effects between occupational factors and psychological outcomes.

Strengthening future studies through probabilistic sampling, repeated-measure designs, and standardized psychometric reporting would considerably enhance the robustness and external validity of research on anxiety and depression in nursing populations.

## 4. Discussion

This review provides an updated synthesis of validated instruments used to assess anxiety and depression among nurses. Given the moderate methodological quality of the evidence included, the following interpretation should be considered cautiously and within the exploratory nature of the available data.

These findings suggest that the reported prevalences and associations may be biased by design; therefore, estimates should be interpreted as indicative rather than conclusive and corroborated by longitudinal designs and complementary objective measures.

The methodological quality of the evidence was generally moderate, with three studies at high risk due to a lack of controls and possible maturation/regression to the mean effects. These results advise caution in interpreting the associations between anxiety/depression and work stressors: although the findings are consistent across multiple contexts, the predominant use of self-reports, convenience sampling, and cross-sectional design may overestimate or underestimate the true magnitudes. Where feasible, future research should incorporate probability sampling, longitudinal or quasi-experimental designs with control groups, and robust confounding control strategies (e.g., propensity score), as well as objective indicators (absenteeism, turnover, workload, incident log) to complement the scales.

Although the review period covered the last ten years, most of the selected studies were published in the last six years, reflecting an upward trend. This pattern coincides with the COVID-19 pandemic and the post-pandemic period, suggesting growing concern in the scientific community about anxiety and depression in nurses.

The present systematic review provides an updated overview of the validated tools used to assess anxiety and depression in nurses, a group particularly vulnerable to these disorders. Among the identified instruments, the PHQ-9 and GAD-7 emerged as the most frequently applied. Their brevity, ease of administration, and sound psychometric support make them practical options for repeated screening in demanding healthcare environments. However, their self-report nature and the limited cross-cultural validation among practicing nurses restrict their interpretability. Likewise, scales such as the GHQ-28 and the BDI provide more detailed assessments, although their length may limit their use in settings with a heavy healthcare load.

These findings are consistent with the international literature, which documents high prevalences of anxiety (23–38%) and depression (25–35%) among nurses and other healthcare professionals during the pandemic, as well as psychosocial stress and occupational violence associated with high levels of stress or burnout [7,36,37]. In particular, almost half of nursing staff reported experiencing physical or verbal abuse related to their role, which has been linked to anxiety–depressive symptoms and chronic emotional exhaustion [38,39,40].

The combined analysis of the included studies suggests that the most widely used tools not only quantify symptoms but also establish associations with contextual factors such as sleep quality, burnout, workplace violence, or resilience. This wealth of data demonstrates the complexity of emotional distress in the nursing context and reinforces the usefulness of a multifactorial approach to its assessment and management. Although the included scales have demonstrated good sensitivity and specificity, it is important to highlight that they are all self-reported, which introduces a subjective component that can limit the objectivity of the assessment [41,42]. It is advisable to integrate objective and contextual indicators—e.g., absenteeism, turnover, workload, incident logs, or sleep metrics—to obtain more valid and comparable estimates.

Although instruments such as the PHQ-9 and GAD-7 have Spanish versions validated in various Spanish-speaking countries [43,44,45], and the GAD-7 has demonstrated good reliability in nursing students, no evidence of formal cross-cultural validation or psychometric processes aimed at practicing nurses has been found. Therefore, it is unknown whether these tools maintain the same psychometric performance in this professional group, which limits their direct applicability and underscores the need to move toward the development of more contextualized instruments.

Finally, the recommendation to implement systematic screening and sustainable psychological-support programs stems from the clear, limited availability of validated assessment tools rather than from empirical evidence on program effectiveness within the included studies. This suggestion aims to guide future initiatives to promote early detection and prevention of anxiety and depression among nurses.

This work provides a comparative mapping of instruments focused on their applicability in nursing, offering practical criteria for selecting the most appropriate tool for the healthcare context.

## 5. Conclusions

The most frequently used tools to assess anxiety and depression in nurses were the PHQ-9, GAD-7, GHQ-28, and BDI, selected for their brevity, reliability, and ease of administration. However, their specific psychometric validation and formal cross-cultural adaptation for Spanish-speaking nursing have not been established, limiting the accuracy with which they can be applied in this professional context. Beyond Spanish-speaking populations, there remains a need to adapt and validate these instruments in other linguistic and cultural settings to ensure broader applicability and comparability.

The recommendation to establish systematic screening and sustainable psychological-support programs derives from the general trends observed in the reviewed studies, rather than from direct empirical evidence of program effectiveness. These measures will contribute to improving early detection, a comprehensive approach to mental health in nursing, and, ultimately, the quality of care and staff retention.

Anxiety and depression in nursing are closely related to work-related stressors; personal resources modulate the response but do not compensate for unfavorable organizational contexts. Given the predominance of self-reports, it is advisable to complement the assessment with objective measures (e.g., absenteeism, turnover, workload, or incidents) whenever feasible. This review offers a useful and applicable synthesis for selecting instruments in nursing settings and emphasizes the importance of future research aimed at achieving cross-cultural validation and methodological refinement in specific populations.

### Limitations

This review did not perform meta-analysis due to the heterogeneity of designs, settings, and metrics. The evidence base was dominated by cross-sectional studies with self-reports and non-probability sampling, which restricts causal inferability and may introduce selection and information bias. Although the JBI checklists were applied, many studies were not designed for psychometric validation in Spanish-speaking nurses; therefore, the assessment of specific measurement properties was beyond the primary scope of this evaluation. Future work is recommended to include country/setting-specific cross-cultural validations and invariance analyses where appropriate.

## Figures and Tables

**Figure 1 ijerph-22-01714-f001:**
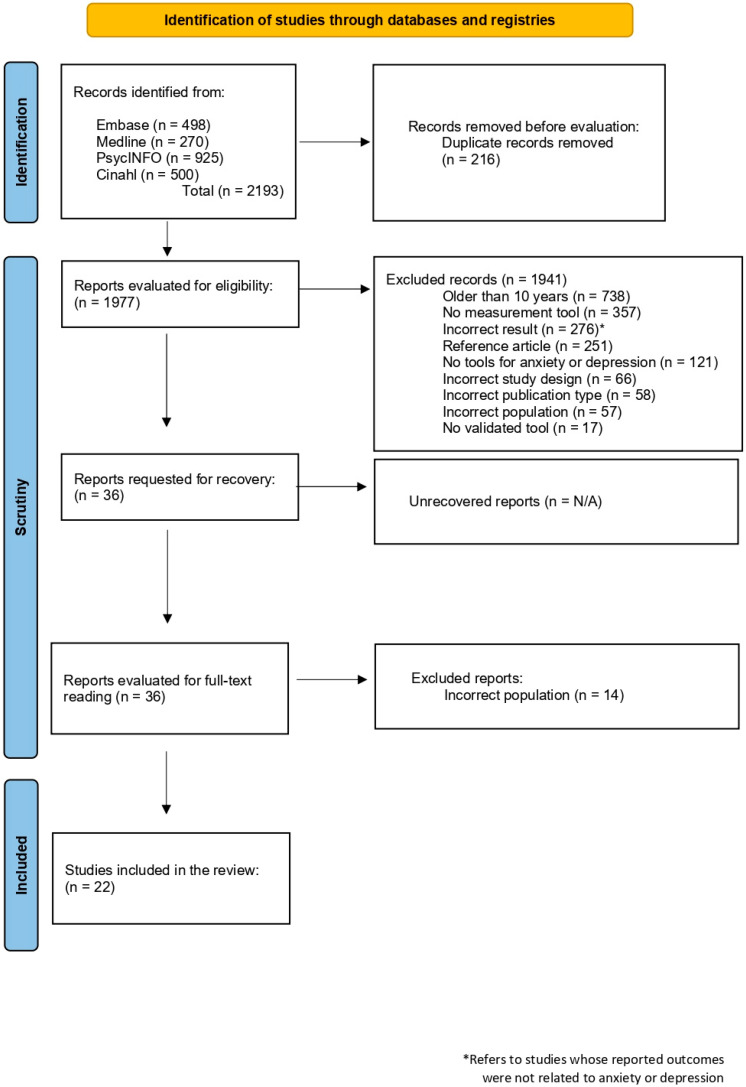
Flowchart of the study selection process according to PRISMA.

**Table 1 ijerph-22-01714-t001:** Characteristics of the studies selected in the review.

First Author/Year	Method	Sample Size	Country	Tools		Results
Chowdhury, S.R. et al. (2022) [14]	Cross-sectional study	1264	Bangladesh	PHQ-9	BMS + WPVS + NAQ + SIJS-5	Depression was positively correlated with violence, harassment, and burnout, and negatively correlated with job satisfaction (*p* < 0.001). Nurses who worked more than 48 h or were not paid on time had higher depression scores.
Potter, G. et al. (2021) [15]	Cross-sectional and prospective study	372	Germany		OLBI	Slower reaction time is associated with burnout but not depression, suggesting distinct cognitive profiles that should be addressed separately in mental health.
Bock, C. et al. (2020) [16]	Cross-sectional study	320	Germany	PHQ-2 + GAD-2	WFZA	Secondary traumatic stress in nurses is associated with higher levels of depression, anxiety, decreased work ability, and less social support, highlighting the need to support their mental health.
Mensinger, J.L. et al. (2022) [17]	Cross-sectional study	467	USA	PHQ -2 + GAD -7	ISI + IES-R	Hospital nurses during COVID-19 showed high levels of distress, with much higher rates of depression and traumatic stress than previous studies, in a context of high mortality among the group.
Ding, C. et al. (2023) [18]	Cross-sectional study	1774	China	PHQ-9 + GAD-7	WVS + SQS + PDQ-5 + MSPSS + CD-RISC-10 + Three item loneliness scale	31.5% of nurses experienced workplace violence, resulting in poor mental health outcomes, underscoring the urgent need for preventative measures.
Marsden, K.M. et al. (2022) [19]	Mixed longitudinal study	1646	Australia		ISI + IES-R	The prevalence of anxiety, depression, insomnia, and PTSD remained high throughout the year. The main predictor of psychological distress was family and household stress, followed by lack of support from the clinical team.
Cook, A. et al. (2021) [20]	Cross-sectional study with convenience sampling	96	USA	GAD-7	MBI	Trauma nurses experience high levels of burnout and anxiety, with a higher risk among those over 50 and those who are white nurses, while family support reduces emotional exhaustion.
Zanjani, M.E. et al. (2021) [21]	Cross-sectional study	200	Australia	GHQ-12	SCAS-R + PSS + NIRTQ-2	Job satisfaction and supportive environments improve psychological health, while sociocultural adaptation reduces stress and improves overall health.
Marin, A.M. et al. (2020) [22]	Descriptive cross-sectional study	210	Spain	GHQ-28 + STAI	DECORE + NASA-TLX	Trait anxiety is associated with mental health in nurses, with high psychosocial risk (86%) and 40% at the emergency level, although the majority showed good mental health.
Kowalczuk, K. et al. (2020) [23]	Cross-sectional study	558	Poland	GHQ-28	SWEQ	Workplace stress affects 83.5% of nurses, with work overload being the key factor, although increased training and responsibility improve mental health.
Łopatkiewicz, A. et al. (2023) [24]	Cross-sectional study	59, 52, 56, 50, 53, 57	Poland, Germany, Italy, Czech Republic, Slovakia, Hungary		MBI	Emotional exhaustion is the leading indicator of poor mental health among psychiatric nursing staff, with the greatest impact on older and more experienced nursing staff. Germans are the most affected, while Slovaks are the least affected.
Seabra, P.R.C. et al. (2019) [25]	Cross-sectional, analytical and observational study with a quantitative approach	1264	Portugal		-	Seventy-six percent of nurses reported significant anxiety, 22.2% had symptoms of major depression, and 94.1% had some form of social dysfunction; better mental health was associated with more sleep and weekends off.
Ariapooran, S. et al. (2018) [26]	Comparative study	303	Iran	BDI	STS + Sexual satisfaction scale	22.4% of married nurses showed STS symptoms, with greater severity than male nurses, and this was associated with more depression and lower sexual satisfaction and marital intimacy.
Esaki, K. et al. (2020) [27]	Longitudinal observational study	683	Japan		SLE’S + LTE + NEO-FF1	CBP significantly reduced depressive symptoms in nurses, especially in those with high or low BDI scores before the intervention, although symptoms worsened at the beginning of work and then stabilized.
Batalla, V. R. D. et al. (2019) [28]	Cross-sectional study	242	the Philippines		SAS + NSS	Spirituality influences depression in nurses, with work-related stress being a moderating factor, highlighting the need for further research within the profession.
Turan, N. et al. (2023) [29]	Intervention study	57	Türkiye		RSA	The training program improved psychological resilience, reduced depression in nurses, and its effects were sustained for two years.
Fadzil, N.A. et al. (2021) [30]	Intervention study	35	Malaysia	DASS-21	PSS	The mindfulness intervention significantly reduced perceived stress and anxiety among nurses.
Delgado, C. et al. (2021) [31]	Cross-sectional study	482	Australia		RAW + PWB	Workplace resilience and postgraduate degrees improved nurses’ psychological well-being, with fewer symptoms of depression, anxiety, and stress than in previous studies, although some participants reported severe mental distress.
Barnett, M.D. et al. (2019) [32]	Cross-sectional study	90	USA		SSS + work-family balance scale	Social support at work reduces psychological distress in palliative care nurses, while satisfaction with work-family balance mediates the relationship with depressive symptoms.
Zhang, L. et al. (2020) [33]	Clinical trial	44	China	HADS	ISI + PSQI	Shimian improved sleep, anxiety, depression, and alertness in on-duty nurses, and changes in salivary cytokines were associated with better outcomes.
Gül, Ş. et al. (2021) [34]	Descriptive cross-sectional study	192	Türkiye	BAI	-	Operating room nurses reported moderate anxiety during COVID-19, associated with chronic illness, fear of infection, and long shifts with insufficient support.
Fu, C. et al. (2023) [35]	Cross-sectional study	1888	China	CES-D-R-10	FFWC scale	Physical violence is associated with a higher prevalence of depressive symptoms (63.7%), while high FFWV increases depression in non-violent groups, highlighting the need for preventive strategies.

**Table 2 ijerph-22-01714-t002:** Strengths and limitations of anxiety and depression measurement tools.

Tool	Advantantages	Disadvantages
PHQ-2	Quick depression screening tool. Easy to administer and assess. It can be used before the PHQ-9 to detect suspected cases.	It does not measure the severity of depression. It can generate false negatives.
PHQ-9	High sensitivity and specificity for detecting depression. Widely validated and used internationally.	It does not distinguish between different types of depression. It may miss atypical or mild symptoms.
GAD-2	Quick anxiety screening tool. Very useful in time-poor settings. It can be used before the GAD-7 to detect suspected cases.	It does not measure the severity of anxiety. It is less accurate than the GAD-7 for diagnosing generalized anxiety.
GAD-7	Quick and reliable for detecting generalized anxiety disorder. Useful for monitoring treatment.	It does not assess other types of anxiety. It may produce false positives in populations with high stress but without anxiety disorders.
GHQ-12	Quicker to administer than the GHQ-28. It assesses mental well-being and general psychological distress. Useful in epidemiological and public health studies.	It does not allow for clinical diagnosis, only initial detection. It may not detect mild or early-stage cases.
GHQ-28	It evaluates depression, anxiety, somatization, and social dysfunction. Good accuracy in detecting general psychological distress.	It does not provide a clinical diagnosis; it only indicates the risk.
BDI	It evaluates the severity of depression in greater depth. Useful in clinical and psychological studies.	Longer and more complex to manage.
DASS-21	It simultaneously assesses depression, anxiety, and stress. Quick and easy to administer.	It does not distinguish between clinical anxiety and situational stress. It may require confirmation with other tools.
HADS	Designed for use in hospital settings. Avoids confusion with somatic symptoms.	It is not useful for assessing disorders beyond mild/moderate anxiety and depression. Less accurate in the general population.
STAI	The difference between anxiety as a state and as a trait. Widely used in psychological research.	It is relatively long (40 items).
BAI	It evaluates anxiety with an emphasis on physical and cognitive symptoms.	It does not distinguish between types of anxiety. It may overestimate anxiety in people with physical symptoms for other reasons.
CES-D-10	Quick and it is used in epidemiological studies. It helps identify depressive symptoms in the general population.	It does not provide a clinical diagnosis. It may not be sufficient to assess severe depression.

**Table 3 ijerph-22-01714-t003:** Quality appraisal (JBI). Compact version for main text.

Study: First Author and Year	Design	Sampling	Confounders Addressed	Overall Risk of Bias	Key Notes/Limitations
Chowdhury, S.R. et al. (2022) [14]	Cross-sectional	Non-probability/convenience (not specified)	Partial	Moderate	cross-sectional design limits causal inference; self-report measures; potential information and common-method bias; non-probability sampling; limited generalizability; context-specific to Bangladesh
Potter, G. et al. (2021) [15]	Observational (cohort/longitudinal)	Non-probability/convenience (not specified)	Partial (longitudinal models)	Moderate	risk of attrition and residual confounding despite adjustment; self-report measures; potential information and common-method bias; non-probability sampling; limited generalizability; context-specific to Germany
Bock, C. et al. (2020) [16]	Cross-sectional	Non-probability/convenience (not specified)	Partial	Moderate	cross-sectional design limits causal inference; self-report measures; potential information and common-method bias; non-probability sampling; limited generalizability; context-specific to Germany
Mensinger, J.L. et al. (2022) [17]	Cross-sectional	Non-probability/convenience (not specified)	Partial	Moderate	cross-sectional design limits causal inference; self-report measures; potential information and common-method bias; non-probability sampling; limited generalizability; context-specific to USA
Ding, C. et al. (2023) [18]	Cross-sectional	Non-probability/convenience (not specified)	Partial	Moderate	cross-sectional design limits causal inference; self-report measures; potential information and common-method bias; non-probability sampling; limited generalizability; context-specific to China; PSM/multivariable adjustment used; residual confounding remains
Marsden, K.M. et al. (2022) [19]	Observational (cohort/longitudinal)	Non-probability/convenience (not specified)	Partial (longitudinal models)	Moderate	risk of attrition and residual confounding despite adjustment; self-report measures; potential information and common-method bias; non-probability sampling; limited generalizability; context-specific to Australia
Cook, A. et al. (2021) [20]	Cross-sectional	Convenience sampling	Partial	Moderate	cross-sectional design limits causal inference; self-report measures; potential information and common-method bias; non-probability sampling; limited generalizability; context-specific to USA
Zanjani, M. E. et al. (2021) [21]	Cross-sectional	Non-probability/convenience (not specified)	Partial	Moderate	cross-sectional design limits causal inference; self-report measures; potential information and common-method bias; non-probability sampling; limited generalizability; context-specific to Australia
Marin, A.M. et al. (2020) [22]	Descriptive cross-sectional	Non-probability/convenience (not specified)	Partial	Moderate	cross-sectional design limits causal inference; self-report measures; potential information and common-method bias; non-probability sampling; limited generalizability; context-specific to Spain
Kowalczuk, K et al. (2020) [23]	Cross-sectional	Non-probability/convenience (not specified)	Partial	Moderate	cross-sectional design limits causal inference; self-report measures; potential information and common-method bias; non-probability sampling; limited generalizability; context-specific to Poland
Łopatkiewicz, A. et al. (2023) [24]	Cross-sectional	Non-probability/convenience (not specified)	Partial	Moderate	cross-sectional design limits causal inference; self-report measures; potential information and common-method bias; non-probability sampling; limited generalizability; context-specific to Poland, Germany, Italy, Czech Republic, Slovakia, Hungary
Seabra, P.R.C. et al. (2019) [25]	Cross-sectional	Non-probability/convenience (not specified)	Partial	Moderate	cross-sectional design limits causal inference; self-report measures; potential information and common-method bias; non-probability sampling; limited generalizability; context-specific to Portugal
Ariapooran, S. et al. (2018) [26]	Comparative cross-sectional	Non-probability/convenience (not specified)	Partial	Moderate	self-report measures; potential information and common-method bias; non-probability sampling; limited generalizability; context-specific to Iran
Esaki, K. et al. (2020) [27]	Observational (cohort/longitudinal)	Non-probability/convenience (not specified)	Partial (longitudinal models)	Moderate	risk of attrition and residual confounding despite adjustment; self-report measures; potential information and common-method bias; non-probability sampling; limited generalizability; context-specific to Japan
Batalla, V.R.D. et al. (2019) [28]	Cross-sectional	Non-probability/convenience (not specified)	Partial	Moderate	cross-sectional design limits causal inference; self-report measures; potential information and common-method bias; non-probability sampling; limited generalizability; context-specific to the Philippines
Turan, N. et al. (2023) [29]	Pre–post/non-randomized intervention	Non-probability/convenience (not specified)	No/minimal	High	non-randomized pre–post without control; expectancy/maturation effects; self-report measures; potential information and common-method bias; non-probability sampling; limited generalizability; context-specific to Türkiye; small, single-arm intervention; limited external validity
Fadzil, N.A. et al. (2021) [30]	Pre–post/non-randomized intervention	Non-probability/convenience (not specified)	No/minimal	High	non-randomized pre–post without control; expectancy/maturation effects; self-report measures; potential information and common-method bias; non-probability sampling; limited generalizability; context-specific to Malaysia; small, single-arm intervention; limited external validity
Delgado, C. et al. (2021) [31]	Cross-sectional	Non-probability/convenience (not specified)	Partial	Moderate	cross-sectional design limits causal inference; self-report measures; potential information and common-method bias; non-probability sampling; limited generalizability; context-specific to Australia
Barnett, M.D. et al. (2019) [32]	Cross-sectional	Non-probability/convenience (not specified)	Partial	Moderate	cross-sectional design limits causal inference; self-report measures; potential information and common-method bias; non-probability sampling; limited generalizability; context-specific to USA
Zhang, L. et al. (2020) [33]	Clinical trial	Non-probability/convenience (not specified)	Partial	Moderate	trial procedures not fully detailed; randomization/blinding unclear; self-report measures; potential information and common-method bias; non-probability sampling; limited generalizability; context-specific to China; biomarker endpoints included, but allocation concealment not reported
Gül, Ş. et al. (2021) [34]	Descriptive cross-sectional	Non-probability/convenience (not specified)	Partial	Moderate	cross-sectional design limits causal inference; self-report measures; potential information and common-method bias; non-probability sampling; limited generalizability; context-specific to Türkiye
Fu, C. et al. (2023) [35]	Cross-sectional	Non-probability/convenience (not specified)	Partial	Moderate	cross-sectional design limits causal inference; self-report measures; potential information and common-method bias; non-probability sampling; limited generalizability; context-specific to China

## Data Availability

Data can be viewed in the OSF database (https://osf.io/r9cmb/?view_only=14d2e77fb7b54062933be1cfd364b80a accessed on 21 February 2025).

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
