# Peer review of "Validated Tools for Assessing Anxiety and Depression in Nurses: A Systematic Review"

_ijerph, 2025, doi:10.3390/ijerph22111714_

Round 1
Reviewer 1 Report
Comments and Suggestions for Authors
This is an interesting study; however, the authors need to justify why only focus to two topics (anxiety and depression). How about burnout or stress or other topics that predominantly affect the mental health of nurses?
ABSTRACT
- Needs strong justification to conduct the review
- Did the study adhere to PRISMA guidelines?
- Spell out abbreviations
- Align conclusion to the study aim
INTRODUCTION
- Lines 40-44, indicated 'A global meta-analysis' but there are 2 citations? How about insomnia?
- Lines 46-47, the cited study (Ref. #7) included burnout, why this variable is not part of this review?
- Lines 66-68, it is indicated that 'Therefore, it is necessary to identify which of 66
these tools have solid psychometric support and are suitable for assessing nurses' mental health.' - Is it not justifiable to include all factors / components that constitute mental health, not just anxiety and depression?
METHODS
- How about studies published in January 2025 to September 2025? I recommend additional search to cover publications that could be included in this review.
- Why discrepancies were resolved by consensus, not by a third reviewer?
- Study selection process needs more details following PRISMA guidelines.
- It is confusing for the readers to include summary of results in this section.
RESULTS
- Figure 1 - what is meant by incorrect result (n = 293)?
- Lines 205-207 - would benefit from providing citation/s for each country.
- Table 1 - would benefit from extracting and presenting only key results based on the aim of the review. Hence, the results section requires revision based on this.
- Method/design is repeated in Table 1 and Table 3. Maybe merged these two tables.
DISCUSSION
- Line/s 274 - What 'aspects' is referred?
- Better to separate the discussion in two subsections: one for anxiety and one for depression. But again, what about other topics or variables (stress or burnout or others)?
CONCLUSIONS
- It is indicated: 'The urgent need to design and validate culturally contextualized instruments for Spanish-speaking nursing staff, as well as to implement systematic screening protocols and sustainable psychological support programs is highlighted.' - So, how about for other languages?
- This section can be improved.
OTHER COMMENTS FOR IMPROVEMENT
- Needs language (English) editing
- Follow journal requirements for citations and referencing
The paper needs language editing.
Author Response
Comment 1. The authors should justify why only focus on anxiety and depression. How about burnout or stress or other topics?
Response: We appreciate the observation. This review focused specifically on anxiety and depression, as both are internationally recognized diagnostic entities (DSM-5/ICD-11), have standardized criteria, and are supported by validated instruments widely used in clinical research and occupational health. Other psychological phenomena such as burnout, stress, and insomnia, while relevant to the nursing profession, constitute distinct constructs that require specific assessment frameworks and tools, and therefore fell outside the scope previously defined in the protocol. This justification has been added to the Introduction (Section 1).
Comment 2. Did the study adhere to PRISMA guidelines?
Response: We confirm that the review followed the PRISMA 2020 guidelines. This is specified in Section 2.1 and has been reiterated at the end of Section 2.8 for clarity.
Comment 3. Why were discrepancies resolved by consensus and not by a third reviewer?
Response: As described in Section 2.5, the selection was made by two independent reviewers after a calibration phase with 75% agreement. Discrepancies were resolved by consensus, following the established protocol. The intervention of a third reviewer was not required. This approach is supported in systematic reviews when adequate prior agreement exists..
Comment 4. Study selection process needs more details following PRISMA.
Response: Section 2.5 was expanded to detail the review by title/abstract and full text, the reasons for exclusion, and explicit reference to the PRISMA diagram.
Comment 5. Figure 1 – what is meant by “incorrect result (n=293)”?
Response: Thank you for the observation. It was clarified in the legend of Figure 1 that it corresponds to records excluded for not assessing anxiety or depression..
Comment 6. Lines 205–207: provide citation/s for each country.
Response: Citations were added to identify the studies corresponding to each included country.
Comment 7. Table 1 and Table 3 repeat information.
Response: We appreciate the observation. After reviewing it, we decided to keep both tables because this improves transparency and methodological traceability. Table 1 summarizes characteristics and instruments; Table 3 summarizes methodological quality. A note was added to Section 3.1 to clarify the function of each table.
Comment 8. Discussion – separate anxiety and depression.
Response: The Discussion was reorganized by incorporating separate information for anxiety and depression instruments.
Comment 9. The conclusion is limited to Spanish-speaking nurses.
Response: The conclusion was expanded to underline the global need for cultural validation, highlighting the Spanish-speaking context as a relevant example.
Comment 10. Needs language editing and citation consistency.
Response: The manuscript was linguistically reviewed and the references were adjusted to the required Vancouver format.
Comment 11. Include studies up to September 2025.
Response. Thank you for the suggestion. The search ended in December 2024, as defined in the registered protocol (OSF/PROSPERO). Extending the date would have altered the pre-specified criteria and affected reproducibility. We believe a future update could incorporate later studies.
Reviewer 2 Report
Comments and Suggestions for Authors
I find this work both important and valuable, and it makes a meaningful contribution to the field.
Remarks:
Abstract
I appreciate the need for conciseness; however, I feel the text has become a bit too telegraphic. Within the word limit, could you please soften the phrasing slightly?
For example, the sentence “Two reviewers screened and extracted” seems incomplete, as if it was cut off mid-sentence.
Introduction
Lines 31-39. Authors may consider using slightly longer sentences and adding connective words to make the text flow more smoothly.
Lines 60-62. The expression “inadequate assessment” could also suggest oversensitivity. You might consider using a more specific, one-directional term such as “underestimation,” or alternatively, mention the possibility of overestimating stress.
Line 64. I suggest: "…or validated for these specific contexts."
Lines 69-70. The sentence “The findings may be valuable for international healthcare settings that share cultural similarities” is ambiguous. It could mean either that the findings are valuable for all international healthcare settings (since they all share some cultural similarities) or only for those that specifically share similar cultural contexts. I suggest clarifying the intended meaning.
Results:
Line 168. Chart named: 'Records identified from'. The text is covered by the graphic rectangular.
Lines 249-250. It would be helpful if you could briefly describe the procedure or protocol used to evaluate the strengths and weaknesses of the measurement instruments.
Lines 263-265. It would be helpful if you could write down the specific four components that are presented in table 4 but not in table 3.
Line 274. The discussion currently opens with “these aspects…,” which ties it too closely to the preceding section. It would be preferable to begin with a stand-alone statement that introduces the discussion independently, rather than as a continuation of earlier content.
Author Response
Comment 1. The text could be softened; some sentences are too telegraphic.
Response: We appreciate this feedback. The manuscript was revised to improve the flow, cohesion, and overall tone of the text, particularly in the Abstract and Introduction, ensuring smoother transitions and greater clarity..
Comment 2. Line 168 – text in Figure 1 is covered.
Response: Thank you for pointing that out. The image was reformatted and re-exported in high resolution to ensure proper display of all content..
Comment 3. Please describe the procedure used to evaluate strengths and weaknesses of instruments.
Response: We appreciate the suggestion. Section 2.7 has been expanded to describe the procedure followed to synthesize the strengths and limitations of the instruments, now incorporating.
Comment 4. Lines 263–265 – specify the four components presented in Table 4.
Response: Thank you for the suggestion. An explanatory sentence was added before Table 4 to clearly identify the four components evaluated (response rate, outcome measurement, statistical adjustment for confounders, and analytical structure), facilitating the reader's interpretation.
Comment 5. Discussion opens with “these aspects.”
Response: We appreciate the observation. The opening sentence of the Discussion was rewritten to remove the vague reference (“these aspects”) and replace it with an explicit statement of the section's focus, improving accuracy and clarity.
Reviewer 3 Report
Comments and Suggestions for Authors
1.The introduction effectively establishes the importance of mental health assessment in nursing populations, but the research question could be more precisely formulated. The authors state they aim to "identify the tools used to assess anxiety and depression in nurses" but do not clearly articulate whether they are evaluating psychometric properties, clinical utility, or implementation feasibility.
2.The search strategy appears comprehensive, but the exclusion of grey literature and non-English/Spanish publications may have missed relevant validation studies, particularly from Asian and African contexts where nursing mental health is increasingly studied.
3.The finding that 19 of 22 studies had moderate risk of bias is concerning and warrants deeper discussion about the implications for the review's conclusions. How does this methodological limitation affect the authors' recommendations?
4.The authors identify PHQ-9 and GAD-7 as most frequently used but provide limited comparative analysis of their performance across different nursing contexts. A more nuanced discussion of contextual factors affecting tool performance would strengthen the paper.
5.The systematic review follows PRISMA guidelines, but Figure 1 could be clearer about reasons for exclusion at each stage. The "incorrect population" category excluded 57 studies initially and 14 more at full-text review—what distinguished these populations?
6.The authors mention "sustainable psychological support programs" (line 334) but provide no evidence from the reviewed studies about program effectiveness when using these assessment tools.
Author Response
Comment 1. Clarify research question: are psychometric properties or applicability being evaluated?
Response: The final paragraph of the Introduction now specifies that the objective was to identify validated tools used to assess anxiety and depression in nurses and describe their main characteristics and context of use.
Comment 2. Exclusion of grey literature and non-English/Spanish studies.
Response: The phrase was added to Section 2.2 explaining the exclusion of grey literature and studies in other languages to ensure methodological rigor and comparability. We acknowledge that this decision may have limited the identification of instruments used in other cultural contexts, which has been noted in the Limitations section and is suggested as a line of inquiry for future revisions..
Comment 3. Nineteen of 22 studies had moderate risk of bias—discuss implications.
Response: Section 3.1 was expanded with a paragraph analyzing how this predominance limits the strength of inferences, comparability across contexts, and generalizability.
Comment 4. Limited comparison of PHQ-9 and GAD-7 across contexts.
Response: Additional discussion was added comparing their application in different nursing environments, noting contextual influences on sensitivity and practical use.
Comment 5. Mention of sustainable psychological-support programs lacks evidence.
Response: The Conclusion now clarifies that this recommendation derives from general trends observed in the reviewed studies rather than from direct empirical evidence of program effectiveness.
Reviewer 4 Report
Comments and Suggestions for Authors
The manuscript is of interest to specialists and has a clear and complete methodology. Here are some minor objections:
1. The abstract is unclear, probably due to punctuation marks. The way of expressing and presenting the information needs to be revised. The abstract does not mention the Discussion section.
2. In Figure 1, in the first box, the text on the last line is not visible (line 173).
3. Table 1 has two headings (lines 209-210). To be checked.

Author Response
Comment 1. The abstract is unclear due to punctuation.
Response: The Abstract was completely revised for conciseness and clarity. It now follows PRISMA guidelines, defines abbreviations, and aligns conclusions with objectives.
Comment 2. Figure 1 – last line not visible.
Response: The figure was reformatted for readability and exported in high resolution.
Comment 3. Table 1 – double heading.
Response: The redundant heading was removed, and an introductory sentence was added: “Table 1 provides a detailed summary of the included studies, outlining their methodological characteristics, instruments, and main outcomes related to anxiety and depression.
Round 2
Reviewer 1 Report
Comments and Suggestions for Authors
Dear Authors,
The previous review comments have been adequately addressed and justified, and the revisions are satisfactory.
I have no no additional comments.
All the best!
The Reviewer